# Inactivated Whole Virus Particle Influenza Vaccine Induces Anti-Neuraminidase Antibodies That May Contribute to Cross-Protection against Heterologous Virus Infection

**DOI:** 10.3390/vaccines10050804

**Published:** 2022-05-19

**Authors:** Chimuka Handabile, Toshiki Sekiya, Naoki Nomura, Marumi Ohno, Tomomi Kawakita, Masashi Shingai, Hiroshi Kida

**Affiliations:** 1International Institute for Zoonosis Control, Hokkaido University, Kita-20 Nishi-10, Kita-ku, Sapporo 001-0020, Japan; chimuka94@czc.hokudai.ac.jp (C.H.); tsekiya@czc.hokudai.ac.jp (T.S.); nomura@czc.hokudai.ac.jp (N.N.); ohnom@czc.hokudai.ac.jp (M.O.); tomomi-k@czc.hokudai.ac.jp (T.K.); shingaim@czc.hokudai.ac.jp (M.S.); 2Global Station for Zoonosis Control, Global Institution for Collaborative Research and Education (GI-CoRE), Hokkaido University, Sapporo 001-0020, Japan; 3Department of Microbiology and Immunology, The Peter Doherty Institute for Infection and Immunity, The University of Melbourne, Melbourne, VIC 3000, Australia; 4Collaborating Research Center for the Control of Infectious Diseases, Nagasaki University, Nagasaki 852-8523, Japan

**Keywords:** inactivated whole virus particle influenza vaccine, anti-neuraminidase antibodies, cross-protection

## Abstract

Despite the use of vaccines, seasonal influenza remains a risk to public health. We previously proposed the inactivated whole virus particle vaccine (WPV) as an alternative to the widely used split vaccine (SV) for the control of seasonal and pandemic influenza based on the superior priming potency of WPV to that of SV. In this study, we further examined and compared the immunological potency of monovalent WPV and SV of A/California/7/2009 (X-179A) (H1N1) pdm09 (CA/09) to generate immune responses against heterologous viruses, A/Singapore/GP1908/2015 (IVR-180) (H1N1) pdm09 (SG/15), and A/duck/Hokkaido/Vac-3/2007 (H5N1) (DH/07) in mice. Following challenge with a lethal dose of heterologous SG/15, lower virus titer in the lungs and milder weight loss were observed in WPV-vaccinated mice than in SV-vaccinated ones. To investigate the factors responsible for the differences in the protective effect against SG/15, the sera of vaccinated mice were analyzed by hemagglutination-inhibition (HI) and neuraminidase-inhibition (NI) assays to evaluate the antibodies induced against viral hemagglutinin (HA) and neuraminidase (NA), respectively. While the two vaccines induced similar levels of HI antibodies against SG/15 after the second vaccination, only WPV-vaccinated mice induced significantly higher titers of NI antibodies against the strain. Furthermore, given the significant elevation of NI antibody titers against DH/07, an H5N1 avian influenza virus, WPV was also demonstrated to induce NA-inhibiting antibodies that recognize NA of divergent strains. This could be explained by the higher conservation of epitopes of NA among strains than for HA. Taking these findings together, NA-specific antibodies induced by WPV may have contributed to better protection from infection with heterologous influenza virus SG/15, compared with SV. The present results indicate that WPV is an effective vaccine for inducing antibodies against both HA and NA of heterologous viruses and may be a useful vaccine to conquer vaccine strain mismatch.

## 1. Introduction

Among pathogens of public health concern are seasonal influenza viruses, which affect over 10% of the world’s population and cause considerable morbidity and mortality [1]. Influenza viruses have two surface spike proteins, hemagglutinin (HA) and neuraminidase (NA), which play roles in the attachment to and release from host cells, respectively, and are also major targets of host immunity. Influenza virus RNA-dependent RNA polymerase has high infidelity due to the lack of proofreading activity during virus replication, leading to the accumulation of mutations under the selection pressure of host immunity, which may cause antigenic variation in viral proteins including HA and NA. This phenomenon is referred to as antigenic drift [2,3,4] and leads to the selection of strains that may not be sufficiently neutralized by pre-existing host immunity induced by either infection or vaccination in the past. Therefore, annual vaccinations are recommended to effectively control seasonal influenza, particularly in high-risk groups such as young children and the elderly [5]. Each year, the World Health Organization holds two meetings to select influenza vaccine strains that antigenically match with strains expected to predominate in the forthcoming season in the northern and southern hemispheres based on global epidemiological surveillance data [5,6,7]. Despite these efforts, vaccine efficacy remains low, ranging from 40% to 70%, and is considerably lower when a mismatch occurs, especially in high-risk populations [8,9,10].

Current influenza vaccines are of two types: live attenuated influenza vaccines and inactivated influenza vaccines (IIV) such as split vaccines (SV), whole virus particle vaccines (WPV), and subunit vaccines [11]. SV consists of viral components that are the products of purified virions that are disrupted with ether or detergent, while subunit vaccine undergoes an additional purification step for the further enrichment of HA or NA. Although SV is the most commonly used vaccine due to its low reactogenicity, it poorly primes naïve individuals [12], which is one of the major problems associated with its use. This feature is considered to have resulted from the disruption of virions, leading to poor activation of immunological cascades after vaccination in naïve populations. On the other hand, WPV is the only IIV in which the virus structure is retained, making it the most immunogenic among contemporary IIVs. Previously, we evaluated the immunogenicity of a single inoculation of WPV and SV in mice and macaques. The results indicated that WPV was superior to SV in priming naïve animals and protecting against homologous virus infection [11,12,13], which reproduces the concern with SV in humans. We also showed that the priming potency of WPV may be linked to its ability to significantly activate the maturation of antigen-presenting cells (APCs) through the efficient delivery of viral RNA into the cells [13]. Because of the high potency in priming, WPV should be used as an alternative vaccine, particularly for naïve populations.

When evaluating influenza vaccines, besides the priming potency, the ability to induce cross-immunity needs to be considered. The widely accepted primary assessment of the efficacy of seasonal influenza vaccines is based on how well they induce antibodies against HA, as determined by the hemagglutination-inhibition (HI) assay [2]. An HI titer of ≥1:40 is believed to correlate with a 50% reduction in the risk of contracting severe disease [14,15,16,17]. However, because host immune responses principally target HA, the HA protein is under higher selective pressure than other viral proteins, resulting in a higher rate of antigenic drift [4,18]. Given the importance of antigenic matching between vaccine strains and prevalent strains, the induction of antibodies against other viral proteins such as more antigenically stable NA may be important to overcome the problem of antigenic drift.

In this study, WPV and SV were evaluated for the induction of cross-reactive immunity utilizing a mouse model. For this purpose, we investigated the protection and antibody responses induced by monovalent WPV or SV of A/California/7/2009 (X-179A) (H1N1) pdm09 (CA/09) against an antigenically drifted strain, A/Singapore/GP1908/2015 (IVR-180) (H1N1) pdm09 (SG/15) (https://www.niid.go.jp/niid/en/influenza-e/2382-flu/flu-antigen-phylogeny/7810-2018-1-29.html (accessed on 10 May 2022)), and an avian strain A/duck/Hokkaido/Vac-3 (H5N1) (DH/07). Both vaccines similarly induced anti-HA antibody responses against CA/09 and SG/15 after the second vaccination. Importantly, WPV of CA/09 successfully induced anti-NA antibodies against CA/09, SG/15, and DH/07 in mice, while SV did not. Given that the rate of antigenic drift is lower in NA than in HA, the present results indicate the potential of WPV to induce effective cross-immunity against mismatched vaccine strains. The use of WPV might reduce the impact of immune evasion not only by antigenically drifted seasonal influenza virus strains but also by pandemic viruses of subtypes related to contemporary influenza virus strains.

## 2. Materials and Methods

### 2.1. Vaccine Preparation

We prepared monovalent WPV and SV from the same batch of purified virus as previously described [19,20]. Briefly, CA/09 was inoculated in 10-day-old embryonated chicken eggs, which were then incubated at 37 °C for 48 h, and chilled at 4 °C overnight, followed by the harvesting of allantoic fluids. Virus particles were concentrated by high-speed centrifugation at 40,000 *g* at 4 °C for 90 min, and then purified by centrifugation at 82,000 *g* at 4 °C for 90 min through a 10–50% sucrose density gradient. The obtained purified virus was divided into two fractions: one fraction was treated with formalin at a final concentration of 0.02% and incubated at 4 °C for 7 days to prepare WPV. To prepare SV, the other fraction was treated with diluted Tween 80 (1:10 ratio with total virus volume; Sigma Aldrich, Burlington, MA, USA) at a final concentration of 0.05% and an equal volume of diethyl ether (Sigma Aldrich, Burlington, MA, USA) and incubated at room temperature for 30 min. After centrifugation at 900 *g* at room temperature for 15 min, the aqueous phase was collected and diethyl ether was evaporated under nitrogen gas. The protein concentrations of both WPV and SV were measured using a NanoDrop One spectrophotometer (Thermo Fisher Scientific, Waltham, MA, USA).

### 2.2. Cells and Viruses

Influenza viruses, CA/09 (ID = EPI_ISL_73798) and SG/15 (ID = EPI_ISL_236221), were kindly provided by the National Institute of Infectious Diseases in Japan. The H5N1 avian strain DH/07 (ID = EPI_ISL_664) was kindly provided by the Laboratory of Microbiology at the Graduate School of Veterinary Medicine, Hokkaido University, Japan. The viruses were propagated in 10-day-old embryonated chicken eggs, and the allantoic fluids were harvested and stored at −80 °C until use. Madin–Darby canine kidney (MDCK) cells were grown in RP10 [consisting of RPMI 1640, (Thermo Fisher Scientific, Waltham, MA, USA), supplemented with 10% fetal bovine serum (FBS; GE Healthcare UK Ltd., Little Chalfont, UK), 50 µM 2-mercaptoethanol (Merck, Darmstadt, Germany), 1 nM sodium pyruvate (Thermo Fisher Scientific), 100 µg/mL streptomycin (Thermo Fisher Scientific, Waltham, MA, USA), 100 U/mL penicillin (Thermo Fisher Scientific), and 20 µg/mL gentamicin (Thermo Fisher Scientific, Waltham, MA, USA)]. The MDCK cells were used for plaque and neutralization assays.

### 2.3. Animals

Eight-week-old female C57BL/6 mice were purchased from Hokudo, Co., Ltd. (Sapporo, Japan), and housed in a BSL-2 animal room at the International Institute for Zoonosis Control under standard laboratory conditions (temperature of 22 °C ± 2 °C, humidity of 50 ± 10%, and a 12 h/12 h light/dark cycle). All mice were given food pellets (CE-2; CLEA Japan, Sapporo, Japan) and water ad libitum.

### 2.4. First Animal Experiment-Vaccination and Virus Challenge Study

Mice were subcutaneously vaccinated twice with either WPV or SV of CA/09 (9 µg protein/100 µL), or PBS as a negative control at a 3-week interval. All vaccinations were performed under inhalation anesthesia with isoflurane (Fujifilm, Wako Pure Chemical Corp., Osaka, Japan). On day 41 from the first vaccination, mice were challenged with a lethal dose of 3000 plaque-forming units (PFU) of SG/15 under the same anesthetic conditions. Daily body weight was measured after the infection with a pre-determined humane endpoint of 25% weight loss relative to weight at the initial measurement. Mice were euthanized by cervical dislocation following anesthesia with isoflurane, at 3- or 5-days post-infection (dpi), and lung samples were collected and homogenized in RPMI_anti_ (RPMI 1640 medium supplemented with 20 µg/mL gentamicin, 100 U/mL penicillin, and 100 µg/mL streptomycin). Following centrifugation of the lung homogenate at 400 g for 5 min, the supernatants were collected and stored at −80 °C until use. The experimental schedule is shown in Figure 1a.

### 2.5. Second Animal Experiment-Vaccination and Serum Collection

Mice were subcutaneously vaccinated three times with either WPV or SV of CA/09 (9 µg protein/100 µL), or PBS as a negative control at 3-week intervals. All vaccinations were performed under inhalation anesthesia with isoflurane. Under the same anesthetic conditions, blood was collected from the tail vein on days 19, 41, and 52 from the first vaccination. After clotting, the supernatants were collected as serum samples and stored at −80 °C until use. The serum samples collected on days 19, 41, and 52 were used for the titration of HI, NI, and neutralizing antibodies against CA/09, SG/15, and DH/07 after vaccination with 1, 2, and 3 doses, respectively. The experimental schedule is shown in Figure 2a.

### 2.6. HI Assay

HI assay was performed as previously described [13]. Briefly, collected serum samples were first treated with a three-fold volume of receptor-destroying enzyme (RDE II; Denka Seiken, Tokyo, Japan) to inactivate nonspecific inhibitors and incubated at 37 °C for 15 h, followed by inactivation of RDE by heating at 56 °C for 60 min. PBS was further added to the samples to dilute the initial serum volume by 10-fold. To carry out HI assay, serial two-fold dilutions of RDE-treated sera were performed in 96-well microplates and mixed with 8 hemagglutination units of the virus antigens (CA/09 or SG/15). The plates were incubated at room temperature for 30 min for antibody-antigen reaction. An equal volume of 0.5% chicken red blood cells was added to the mixtures and further incubated for 30 min. The HI antibody titer was determined as the reciprocal of the highest dilution that inhibited hemagglutination.

### 2.7. Neuraminidase and NI Assays

The neuraminidase assay (NA assay) was used to determine the optimal virus concentration for the NI assay. The NA and NI assays described by Aymard-Henry et al. [21] were performed with modifications as follows. For the NA assay, 10 µL of virus stock solution (CA/09/, SG/15, or DH/07) was serially diluted two-fold with PBS in Eppendorf tubes and 20 µL of fetuin substrate (containing 2.5% (*w/v*) fetuin; Sigma Aldrich, and phosphate buffer (0.4 M Na_2_HPO_4_ and 0.4 M NaH_2_PO_4_); Kanto Chemical Co., Inc., Tokyo, Japan) was added to all tubes. Following incubation at 37°C with 5% CO_2_ for 16 h, 20 µL of periodate reagent (comprising 4.48% (*w/v*) NaIO_4_ and 62% 14 M H_3_PO_4_); Kanto Chemical Co., Inc., Tokyo, Japan) was added. The tubes were incubated at room temperature for 20 min followed by the addition of arsenite reagent (comprising 10% (*w/v*) NaAsO_2_, 7.1% (*w/v*) Na_2_SO_4_, and 0.3% concentrated H_2_SO_4_; Sigma Aldrich, Burlington, MA, USA) at ten times the volume of periodate reagent, and the tubes were vortexed thoroughly until the brown color disappeared. Then, 500 µL of thiobarbituric acid reagent (comprising 0.6% (*w/v*) C_4_H_4_N_2_O_2_S and 7.1% (*w/v*) Na_2_SO_4_; Kanto Chemical Co., Inc., Tokyo, Japan) was added. Reaction tubes were incubated in boiling water (100 °C) for 15 min. After cooling at room temperature, 600 µL of butanol reagent (containing 95% C_4_H_10_OH and 5% 12 M HCl; Sigma Aldrich, Burlington, MA, USA) was added. The tubes were centrifuged for 10 min at 1600 g, after which 100 µL of the pink chromophore in the butanol phase from each tube was transferred to a transparent flat-bottomed 96-well plate, and the optical density (OD) was measured at 540 nm using an iMark microplate reader (Bio-Rad, Hercules, CA, USA). The virus dilution giving an OD reading of 0.5 was used for the subsequent NI assay.

To perform the NI assay, 10 µL serum samples were serially diluted two-fold with PBS in Eppendorf tubes and equal volumes of diluted virus solutions (as determined by NA assay) were added to each tube. The tubes were incubated at room temperature for 60 min, after which 20 µL of fetuin substrate was added, followed by incubation at 37 °C with 5% CO_2_ for 16 h. The NA assay protocol was followed for the remaining NI assay steps. The OD was plotted against the sample dilution, and the NI antibody titer was determined as the reciprocal of the highest dilution with ≤50% OD obtained in the control tubes containing PBS instead of serum.

### 2.8. Neutralization of Viral Infectivity Test

Neutralization of viral infectivity test was performed to measure neutralizing antibodies in the serum. A monolayer of MDCK cells was incubated at 37 °C and 5% CO_2_ overnight in six-well plates at a density of 4.75 × 10^6^ in 3 mL of RP10. The RDE-treated sera prepared as described above were serially diluted two-fold in PBS in a 96-well plate and incubated with 100 PFU of the virus antigens (CA/09 or SG/15) for antigen–antibody reaction. After 60 min, the mixture of virus and serum was added to MDCK cells and incubated at 37 °C and 5% CO_2_ for 45 min with shaking every 15 min to allow the non-neutralized virus to adsorb to the cells. Warmed L15 overlay medium (consisting of Leibovitz L-15 (Life Technologies Corp., Carlsbad, NY, USA) with glutamine at pH 6.8 supplemented with 100 U/mL penicillin, 100 mg/mL streptomycin, 0.028% (*w/v*) NaHCO_3_, 1 mg/mL TPCK-treated trypsin (Worthington Biochemical Corp., Lakewood, NJ, USA), and 0.9% (*w/v*) agarose) was added to the six-well plates. The plates were incubated at 37 °C with 5% CO_2_ for 3 days. Following manual counting of the plaques, the neutralizing antibody titer was determined as the reciprocal of the highest dilution that prevented the growth of plaques to 50% of that obtained in the control wells.

### 2.9. Plaque Assay

A monolayer of MDCK cells was incubated at 37°C with 5% CO_2_ overnight in six-well plates at a density of 4.75 × 10^6^ in 3 mL of RP10. The RP10 medium was aspirated, the cell monolayers were washed with RPMI_anti_, and 125 µL of diluted lung homogenate (10^−1^ to 10^−3^) in RPMI_anti_ was added to each well, except for the control plate. The plates were placed in an incubator at 37 °C with 5% CO_2_ for 45 min with shaking every 15 min to allow virus attachment, followed by the addition of warmed L15 overlay medium to each six-well plate. The plates were incubated at 37 °C and 5% CO_2_ for 3 days. The plaques were manually counted following plaque formation and the lung virus titers were determined.

### 2.10. Ethics Statement

All animal experiments were performed under approval (approval number 17-0003) from the Animal Care and Use Committee of Hokkaido University following Fundamental Guidelines for the Proper Conduct of Animal Experiment and Related Activities in Academic Research Institutions under the jurisdiction of the Ministry of Education, Culture, Sports, Science and Technology in Japan.

### 2.11. Statistical Analysis

All statistical analyses were performed using GraphPad Prism 8 software (GraphPad Software, San Diego, CA, USA). Significant differences were denoted by *p* values < 0.05 using one-way analysis of variance (ANOVA) with multiple comparisons (Tukey’s multiple-comparison test for parametric one-way ANOVA or Kruskal–Wallis test for non-parametric one-way ANOVA).

## 3. Results

### 3.1. Mice Vaccinated with WPV Exhibited Superior Cross-Protection Compared with Those Vaccinated with SV

We compared the protective potency between WPV and SV in mice against heterologous influenza virus SG/15, one of the drifted strains from the vaccine strain CA/09. Three weeks after two subcutaneous vaccinations with either WPV or SV of CA/09 or PBS as a control, mice were challenged with 3000 PFU of SG/15 (Figure 1a) and monitored daily for changes in body weight (Figure 1b). After infection, mice inoculated with PBS (control group) showed signs of decreased activity and/or a hunched back and over 25% loss of their body weight, reaching the humane endpoint at 5 dpi. Mice vaccinated with SV showed similar clinical symptoms, but mean body weight loss was only approximately 10%. In contrast, mice immunized with WPV demonstrated negligible weight loss and improved activity compared with either PBS- or SV-inoculated group. Virus titers in the lungs at 3 and 5 dpi were determined by the plaque assay (Figure 1c,d, respectively). At 3 dpi, mice in the control group had the highest virus titers in the lungs of over 10^6^ PFU, and SV-vaccinated mice had virus titers of around 10^5^ PFU. Meanwhile, three out of five WPV-vaccinated mice had undetectable virus titers, and two other mice also had significantly low titers of virus in the lungs compared with mice inoculated with PBS or SV (Figure 1c). At 5 dpi, the virus titers in the lungs of SV- and WPV-vaccinated mice were even lower than at 3 dpi (Figure 1d). Although the difference did not reach significance, virus titers in the lungs at 5 dpi tended to be lower in the WPV group than in the SV group. Based on these results, WPV appears to be more effective than SV to protect against heterologous virus infection.


Figure 1Protectivity of WPV and SV against heterologous virus challenge. (**a**) Schematic representation of the experimental schedule. C57BL/6 mice were subcutaneously inoculated with PBS or 9 µg of protein of monovalent CA/09 WPV or SV twice at 3-week intervals. Following the second vaccination, mice were challenged intranasally with 3000 PFU of SG/15. (**b**) The change of body weight of mice was calculated as a percentage of the original weight at 0 days post-infection (dpi). Five mice in each group were euthanized for the collection of lung samples at 3 and 5 dpi. (**c**,**d**) Plaque assays were performed to calculate the lung viral titers at 3 (**c**) and 5 dpi (**d**). Samples that had undetectable plaques were assigned a value of 10 for graphing purposes. Dots represent individual values, and lines indicate the mean values with SEM (*n* = 10 per group). (**a**–**c**) The circles, squares, and triangles indicate data from mice inoculated with PBS, SV, and WPV, respectively. * *p* < 0.05, ** *p* < 0.01, *** *p* < 0.0005, **** *p* < 0.0001, one-way ANOVA using a multiple-comparison correction; ns indicates not significant (*p* > 0.05).
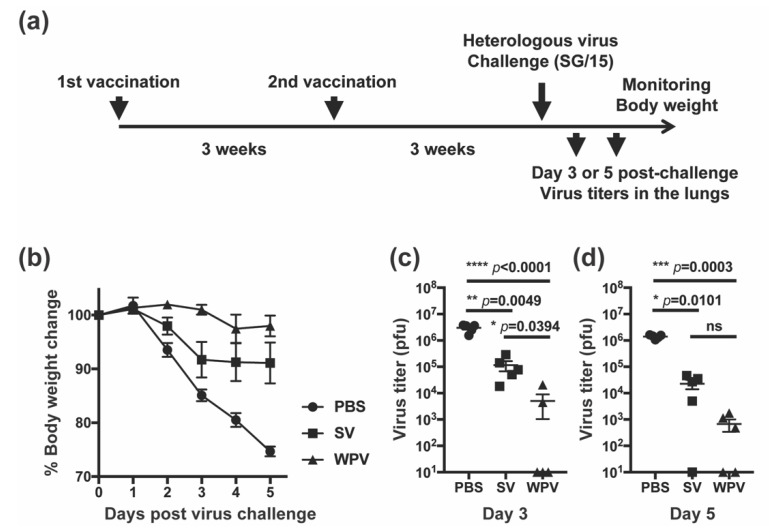



### 3.2. After Multiple Vaccinations, Mice Inoculated with SV Induced HI Antibodies Comparable to Those of Mice Vaccinated with WPV

Next, we investigated the induction of humoral immunity by WPV and SV. Here, mice were given an additional third dose to evaluate how both vaccines boost antibodies against HA and NA. Serum samples were serially collected from the same mice after one, two, and three doses of WPV or SV and were evaluated for the induction of HI antibody titers against homologous (CA/09) and heterologous (SG/15) strains (Figure 2a). Three weeks after the first vaccination (primary response), mice vaccinated with WPV showed significantly higher titers of HI antibodies against both CA/09 and SG/15 than mice inoculated with PBS or SV (Figure 2b,e, respectively). While a significant difference in HI antibody titers against CA/09 after the second vaccination was observed between SV and WPV groups (Figure 2c), those against SG/15 were not significantly different between the groups (Figure 2f). Respective HI antibody titers in WPV- and SV-vaccinated animals reached comparable values against both CA/09 and SG/15 after the third and second vaccinations (Figure 2d,g, respectively). These results imply that both WPV and SV have the potential to induce HI antibodies against the homologous and heterologous strains tested here. Furthermore, given similar HI antibody titers against SG/15 in both vaccine groups after the second vaccination, it indicated that there was no significant contribution of HI antibodies to better cross-protection against the heterologous virus challenge in WPV-vaccinated mice shown in Figure 1.


Figure 2HI antibody titers after vaccination with WPV or SV. (**a**) Schematic representation of the experimental schedule. C57BL/6 mice were subcutaneously vaccinated three times at 3-week intervals with 9 µg of protein of monovalent CA/09 WPV or SV. HI antibody titers against homologous CA/09 (**b**–**d**) and heterologous SG/15 (**e**–**g**) were evaluated in serum samples collected on days 19 (**b**,**e**), 41 (**c**,**f**), and 52 (**d**,**g**). Dots represent individual values, and the mean values with SEM are indicated by lines (*n* = 5 per group). The circles, squares, and triangles indicate data from mice inoculated with PBS, SV, and WPV, respectively. * *p* < 0.05, ** *p* < 0.01, **** *p* < 0.0001, one-way ANOVA using a multiple-comparison correction; ns indicates not significant (*p* > 0.05).3.2. After Multiple Vaccinations, Mice Inoculated with SV Induced HI Antibodies Comparable to Those of Mice Vaccinated with WPV.
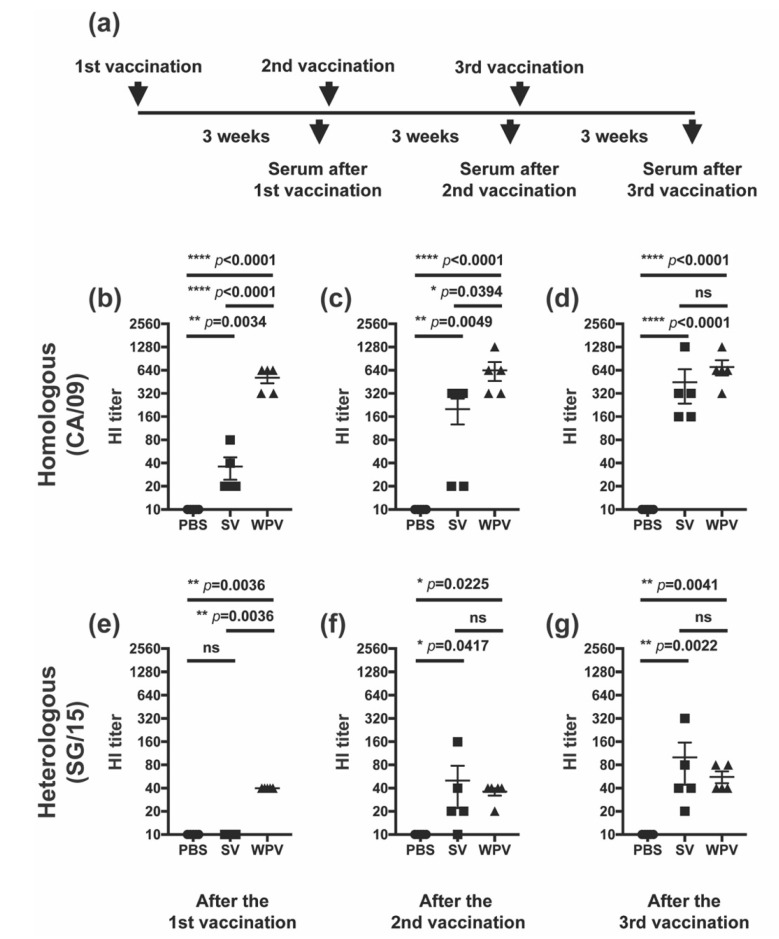



### 3.3. WPV Induced Better NI Antibodies than SV

In addition to anti-HA antibodies, we further evaluated WPV and SV for the induction of antibodies against NA, the second major surface glycoprotein of influenza viruses. Serum samples collected from mice vaccinated with either WPV or SV as described above were subjected to the NI assay. NI antibody titers against homologous CA/09 in the WPV-vaccinated group were significantly increased even after the first vaccination, and the values were further elevated by additional vaccinations (Figure 3a–c). On the other hand, NI antibody titers against CA/09 in SV-vaccinated mice ranged from undetectable to very low at all-time points. Even after the third vaccination, two out of five mice produced no detectable NI antibodies (Figure 3c).

We also evaluated the serum titers of the anti-NA antibodies against heterologous strain SG/15 with an NA amino acid sequence identity of 96% (SG/15 NA; Accession No. EPI849450) compared to that of homologous strain CA/09 (CA/09 NA; Accession No. EPI254042,). Although a single inoculation of WPV significantly but only slightly increased the NI antibody titers against SG/15 (Figure 3d), the titers were significantly elevated by the second and third shots of WPV (Figure 3e,f, respectively). The NI antibody titers against SG/15 in SV-vaccinated mice were below the detectable limit at almost all-time points, as in the PBS control group (Figure 3d–f).

We further examined whether either vaccine effectively induces NI antibodies against H5N1 avian influenza virus strain DH/07 with 90% NA amino acid sequence identity compared to that of CA/09 (DH/07 NA; Accession No. EPI3498). Mice vaccinated with SV did not produce detectable NI antibodies after one or multiple vaccinations (Figure 3g–i). Interestingly, NI antibodies against DH/07 were detectable in the WPV-vaccinated mice even after the first inoculation and increased with subsequent doses, reaching a titer of 71 ± 14 after the third shot. The NI antibody titers in the WPV group were significantly higher than those in PBS- or SV-injected groups at all-time points. These findings suggest that WPV has excellent ability to induce effective NI antibodies against a wide range of viral strains, which may be one of the features that distinguishes WPV from SV.


Figure 3NI antibody titers after vaccination with WPV or SV. C57BL/6 mice were subcutaneously vaccinated three times at 21-day intervals with 9 µg of protein of monovalent CA/09 WPV or SV, as shown in Figure 2a. NI antibody titers against homologous CA/09 (**a**–**c**) and heterologous SG/15 (**d**–**f**) and DH/07 (**g**–**i**) were evaluated in serum samples collected on days 19 (**a**,**d**,**g**), 41 (**b**,**e**,**h**), and 52 (**c**,**f**,**i**). Dots represent individual values, and the mean values with SEM are indicated by lines (*n* = 5 per group). The circles, squares, and triangles indicate data from mice inoculated with PBS, SV, and WPV, respectively. * *p* < 0.05, ** *p* < 0.01, *** *p* < 0.001, **** *p* < 0.0001, one-way ANOVA using a multiple-comparison correction; ns indicates not significant (*p* > 0.05).
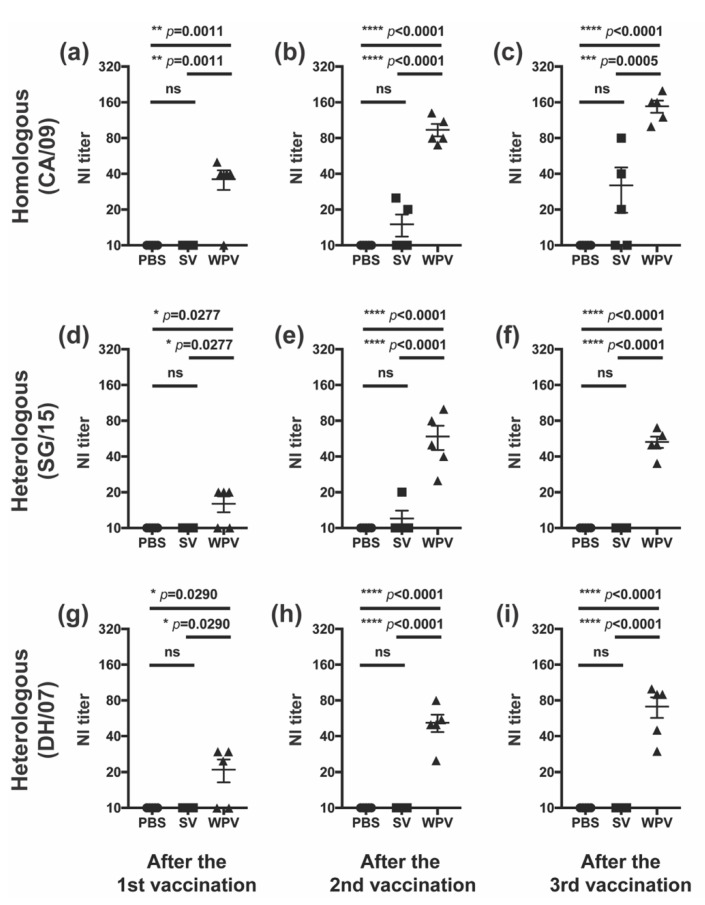



In addition to HI and NI antibody titers, neutralizing antibody titers were also evaluated in serum samples collected 3 weeks after the second inoculation with WPV or SV. Neutralizing antibody titers against homologous (CA/09) or heterologous (SG/15) strains were induced by both WPV and SV, but the titers against both strains were significantly higher in WPV-vaccinated mice than those induced by SV (Figure 4a,b). Taken together with the results of HI and NI antibody titers, the higher titer of neutralizing antibodies observed in WPV-vaccinated mice may be attributed to the superior inducibility of NI antibodies by WPV. Collectively, the findings of the present study suggest that WPV confers better cross-protective immunity against heterologous virus infection than SV.


Figure 4Neutralizing antibody titers after vaccination with WPV and SV. C57BL/6 mice were subcutaneously vaccinated twice at 3-week intervals with 9 µg of protein of monovalent CA/09 WPV or SV, as shown in Figure 1a. Serum samples collected on day 41 were evaluated for neutralizing antibody titers against CA/09 (**a**) and SG/15 (**b**). The circles, squares, and triangles indicate data from mice inoculated with PBS, SV, and WPV, respectively. * *p* < 0.05, ** *p* < 0.01, **** *p* < 0.0001, one-way ANOVA using a multiple-comparison correction.
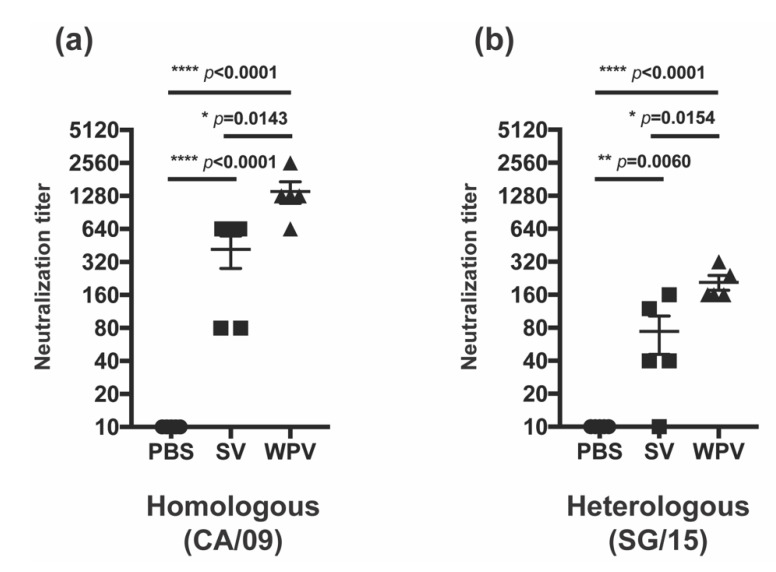



## 4. Discussion

Here, we investigated the protectivity of mice from heterologous virus infection and antibody induction against homologous and heterologous viruses by WPV and SV of CA/09 in mice. In the challenge experiment with the heterologous virus, SG/15, mice vaccinated with WPV had significantly reduced virus titers in the lung, milder body weight loss, and improved activity than those receiving SV (Figure 1b–d), suggesting a superior preventive effect of WPV against severe disease caused by vaccine-mismatched strains. Although these differences in protectivity were observed between WPV and SV, antibodies against HA of SG/15 were comparable after the second vaccination between mice receiving either vaccine (Figure 2f,g). In contrast to anti-HA antibodies, animals inoculated with WPV had higher antibody titers against NA of the heterologous virus SG/15 as well as homologous strain CA/09, which may explain the higher titers of neutralizing antibody against both strains in WPV-vaccinated mice (Figure 3d–f). Furthermore, anti-NA antibodies induced by WPV reacted against the avian influenza H5N1 strain, DH/07 (Figure 3g–i). While numerous studies have been published on anti-NA antibodies and their benefits to individuals during infection [22,23,24,25,26,27,28], few have compared SV and WPV for their induction of NA antibodies against heterologous and avian influenza viruses. The results presented here highlight one of the disadvantages of the widely used seasonal influenza vaccines, SV, and we believe this knowledge can be a useful guide for the selection of vaccine strain. Vaccination with WPV may have greater potential than SV to protect against infection with heterologous virus strains, in part due to NA antibodies.

The primary assessment of the efficacy of influenza vaccines is based on the induction of anti-HA antibodies, as determined using the HI assay. Based on this assay, both SV and WPV showed potency in inducing anti-HA antibodies, achieving similar levels after two or three inoculations in this study. Interestingly, despite inducing similar levels of HI antibodies against SG/15 after the second vaccination, following virus challenge, the lung virus titers were significantly lower in the WPV-inoculated group at 3 dpi than in the SV-vaccinated group. Reports have been published describing that the suppression of viral growth is associated with the induction of anti-NA antibodies [29,30], whose functionality is similar to NA inhibitors, the main antivirals for the therapeutic treatment of influenza. NA inhibitors and anti-NA antibodies prevent virus budding and spread from infected cells, thus suppressing viral growth, particularly during the early stages of infection. Taken together with previous reports, the anti-NA antibodies induced by WPV observed in this study could have contributed to the additional suppression of viral growth, compared with SV. Therefore, understanding of vaccine efficacy could be improved by incorporating analyses involving antibodies against other viral proteins such as NA, in addition to the HI assay.

Several factors may account for the low or negligible induction of anti-NA antibodies by SV. First, ether- and detergent-disrupted SVs are poor immunogens in mice and humans [31,32] because the viral RNA in the SVs, which affects APC activation, is degraded [13]. Second, the HA protein, which binds to sialic acid receptors, is preferentially endocytosed by APCs over NA or other viral proteins, subsequently inducing antibodies mainly against HA. In WPV, the intact virus structure provides an advantage during antigen presentation as the whole virus particle, together with all other virus proteins, can be taken up and processed by APCs, yielding a broader repertoire of antibodies. In the present study, we found that anti-NA antibodies induced by WPV were reactive even to an avian H5N1 virus strain. The ability of WPV to induce both anti-HA and NA antibodies not solely against the homologous virus but also against the heterologous viruses makes its use advantageous for controlling seasonal influenza caused by antigenic variant viruses.

Historically, pre-existing NA antibodies within populations that cross-react with novel strains have been associated with reduced severity of disease, and this was the case during the 2009 and 1968 pandemics caused by H1N1 and H3N2 strains, respectively [33,34,35]. In line with this concept, the use of WPV as a seasonal influenza vaccine strain may partially contribute to the control of pandemic influenza caused by the H5N1 strain via N1 NA antibodies.

Vaccination for the control of seasonal influenza is clearly fundamental for reducing severe morbidity and mortality. A vast number of studies on the development of immunogenic, cross-reactive, and safe vaccines are ongoing. Our series of studies has shown that WPV is not only an excellent vaccine for naïve populations because of efficient priming, but also has the remarkable advantage of protecting against heterologous virus strains. It also induces a broad range of anti-NA antibodies that inhibit NAs of heterologous strains, which may protect against infection with divergent strains. We hereby propose that WPV is ideal not only for the control of seasonal influenza but also for establishing preparedness for future pandemics.

## Data Availability

Publicly available datasets were analyzed in this study. This data can be found here [https://www.niid.go.jp/niid/en/influenza-e/2382-flu/flu-antigen-phylogeny/7810-2018-1-29.html, SG/15 NA; Accession No. EPI849450, CA/09 NA; Accession No. EPI254042), DH/07 NA; Accession No. EPI3498)] (accessed on 10 May 2022).

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
