# Peer review of "Inactivated Whole Virus Particle Influenza Vaccine Induces Anti-Neuraminidase Antibodies That May Contribute to Cross-Protection against Heterologous Virus Infection"

_vaccines, 2022, doi:10.3390/vaccines10050804_

Round 1
Reviewer 1 Report
In this manuscript, researchers compared the immunogenicity and protective potential of split (SV) and whole virus particle vaccine (WPV) in a mouse model of influenza virus vaccination/challenge. They looked for antibody responses against two heterologous viruses while protection was tested against one. Comments to improve manuscript are as follows:
- In the title, authors need to change 'cross-protection against heterologous viruses' to 'cross-protection against heterologous virus', as they tested protective efficacy only against one heterologous virus. Further, based on the results and current experimentation, authors can only hypothesize that the better protection observed in WPV is due to anti-neuraminidase antibodies. They do not have any confirmatory experiments, and hence the title should be changed as 'anti-neuraminidase antibodies that may contribute' (add may).
- Body mass change (Figure 1b) need to be evaluated using repeated measures ANOVA and show whether there is statistical difference or not. Besides, area under the curve can also be used to compare changes in body mass post-challenge.
- What is the detection limit for virus titers? Include that.
- Include the neutralizing antibody titers against the avian virus used for HI and NI titers.
Author Response
Thank you for your kind help for revising manuscript entitled “Inactivated Whole Virus Particle Influenza Vaccine Induces Anti-Neuraminidase Antibodies That May Contribute to Cross-protection Against Heterologous Virus Infection” to [Vaccines]. We are grateful to you for the time dedicated to reviewing the original version of the manuscript, for the comments, and for the insightful suggestions. We have incorporated most of your suggestions.
The below shows the point-by-point response to comments and concerns.
- In the title, authors need to change 'cross-protection against heterologous viruses' to 'cross-protection against heterologous virus',as they tested protective efficacy only against one heterologous virus.Further, based on the results and current experimentation, authors can
only hypothesize that the better protection observed in WPV is due toanti-neuraminidase antibodies. They do not have any confirmatory experiments, and hence the title should be changed as 'anti-neuraminidase antibodies that may contribute' (add may).
Response: According to your comment, the title of the manuscript has been changed to “Inactivated Whole Virus Particle Influenza Vaccine Induces Anti-Neuraminidase Antibodies That May Contribute to Cross-protection Against Heterologous Virus Infection”
- Body mass change (Figure 1b) need to be evaluated using repeated measures ANOVA and show whether there is statistical difference or not.
Response: Significant difference was observed between WPV and PBS using repeated-measures ANOVA, but not between WPV and SV.
- What is the detection limit for virus titers? Include that.
Response: For samples that did not have detectable plaques, a value of “10” was assigned for graphing purposes. We have added this information in the figure legend (Line 273-274).
Include the neutralizing antibody titers against the avian virus used for HI and NI titers
Response: Although neutralization assay was performed for the avian influenza virus, titers were below the detection limit thus the data was not included.
Reviewer 2 Report
In the current study Handabile et al revisit how influenza A virus (IAV) anti-Neuraminidase antibody responses contribute to protection against IAV. IAV remains a global health concern and means to provide universal protection through vaccination are needed. The study is well executed and the manuscript well-written.
Concerns regarding the manuscript in its current form are as follows:
- No experimental evidence is provided to show that it is anti-neuraminidase antibody that is providing cross-protection. While it is detected and correlates with protection, definitive data showing that it alone is responsible is needed to support the major claims and title of the study.
- How the current study is a significant advance over the wealth of literature studying anti-Neuraminidase immunity against IAV is underdressed within the manuscript and needs to be discussed.
- The rationale for switching from boosting with vaccine formulations twice to three times for serological assessments should be provided.
- Only female animals were used in the study and thus the study does not address the important biological variable of sex that could impact study outcomes. Experiments with male mice are needed.
- While it is well established and accepted what HAI correlate with 50% protection, it is unclear what value and whether NI antibody titers similarly can be used as indicators. This is a major weakness that needs to be addressed.
Author Response
Thank you for your kind help for revising manuscript entitled “Inactivated Whole Virus Particle Influenza Vaccine Induces Anti-Neuraminidase Antibodies That May Contribute to Cross-protection Against Heterologous Virus Infection” to [Vaccines]. We are grateful to you for the time dedicated to reviewing the original version of the manuscript, for the comments, and for the insightful suggestions. We have incorporated most of the your suggestions.
The below shows the point-by-point response to comments and concerns.
- No experimental evidence is provided to show that it is anti-neuraminidase antibody that is providing cross-protection. While it is detected and correlates with protection, definitive data showing that it alone is responsible is needed to support the major claims and title of the study.
Response: Yes, we agree with your comment. Our current study only speculates the contribution of NA antibodies to improved protection by WPV. Thus, we have edited the title of this study. We are now conducting further experiments to elucidate the contribution of NA antibodies to the protection including challenge infection with HPAIV H5N1 and the use of recombinant NA as a vaccine antigen.
How the current study is a significant advance over the wealth of literature studying anti-Neuraminidase immunity against IAV is underdressed within the manuscript and needs to be discussed.
Response: We agree with your comment. We have added a few sentences in the discussion addressing your concern (Line 385-390). We are conducting further experiments to investigate anti-NA immunity and its contribution to protection against influenza virus infection.
The rationale for switching from boosting with vaccine formulations twice to three times for serological assessments should be provided.
Response: Thank you. In the second experiment for serological assessments, we aimed at investigating how an additional dose of either vaccine boosts antibodies against HA and NA. We have added a sentence in the results section addressing this point (Line 281-284).
- Only female animals were used in the study and thus the study does not address the important biological variable of sex that could impact study outcomes. Experiments with male mice are needed.
Response: Thank you for your comment. It is true that immune responses between male and female mice are different and conducting experiments in both sexes would result in better inferences. We hope to investigate this point in our future studies. However, our previous studies with WPV and SV were also conducted in female mice only (referenced in 12 and 13) and the results agreed with those obtained in female and male macaques. Thus, we used female mice for the current study.
While it is well established and accepted what HAI correlate with 50% protection, it is unclear what value and whether NI antibody titers similarly can be used as indicators. This is a major weakness that needs to be addressed.
Response: We agree with your comments. Investigating the value of NI antibody titers that corelate with protection would be a significant advancement in anti NA-studies, however, it is beyond the scope of the current study. Since the study is still ongoing, we hope to address this point in the future.
Reviewer 3 Report
Dear authors
I hope all of you are fine. Regarding the revision of the Manuscript (vaccines-1700882), titled “Inactivated Whole Virus Particle Influenza Vaccine Induces Anti-Neuraminidase Antibodies That contribute to Cross-protection Against Heterologous Viruses”. Really it is an interesting research discussing the cross protection of an inactivated WVP vaccine candidate.
However, some comments should be replied.
- In materials and methods (M & M): please add more details for the used viruses in both experiments (Accession No., origin)
- Please indicate the antigenic drift of A/Singapore/GP1908/2015 (IVR-180) (H1N1) pdm09 (SG/15) compared to A/California/7/2009 (X-179) (H1N1) pdm09 (CA/09)].
- Do you have any data (e.g. phylogenetic analysis) confirming the antigenic similarity between NA of A/California/7/2009 (X-179) (H1N1) pdm09 (CA/09) and A/duck/Hokkaido/Vac-3/2007 (H5N1) (DH/07), please add it or its refernce.
- In M & M, line 136: please add the title of (First experiment- Vaccination challenge study).
- In M & M, line 150: please add the title of (Second experiment- Vaccination study).
- I didn’t see a negative control mice groups (1st study: non-vaccinated, non-challenged) (2nd study: non-vaccinated control), if you have these data, please add it.

Author Response
Thank you for your kind help for revising manuscript entitled “Inactivated Whole Virus Particle Influenza Vaccine Induces Anti-Neuraminidase Antibodies That May Contribute to Cross-protection Against Heterologous Virus Infection” to [Vaccines]. We are grateful to you for the time dedicated to reviewing the original version of the manuscript, for the comments, and for the insightful suggestions. We have incorporated most of the your suggestions.
The below shows the point-by-point response to comments and concerns.
- In materials and methods (M & M): please add more details for the used viruses in both experiments (Accession No., origin)
Response: Thank you. We briefly mentioned the sources of the viruses in the acknowledgments and have included the identity numbers of all viruses from GISAID (Line 122 and 124).
- Please indicate the antigenic drift of A/Singapore/GP1908/2015 (IVR-180) (H1N1) pdm09 (SG/15) compared to A/California/7/2009 (X-179A) (H1N1) pdm09 (CA/09)].
Response: Thank you for good suggestion. A link from the National Institute of Infectious Diseases (NIID) in Japan, showing the antigenic drift of A/Singapore/GP1908/2015 (IVR-180) (H1N1) pdm09 has been added (Line 94-95). A/California/7/2009 (X-179) (H1N1)pdm09 is the original pandemic strain and A/Singapore/GP1908/2015 (IVR-180) (H1N1) pdm09 is a derivative strain belonging to 6B.1 clade. A/Singapore/GP1908/2015 (IVR-180) (H1N1) pdm09 replaced A/California/7/2009 (X-179) (H1N1)pdm09 as a vaccine antigen in the 2017/2018 flu season in Japan, because of antigenic difference.
- Do you have any data (e.g. phylogenetic analysis) confirming the antigenic similarity between NA of A/California/7/2009 (X-179) (H1N1)pdm09 (CA/09) and A/duck/Hokkaido/Vac-3/2007 (H5N1) (DH/07), please add it or its reference.
Response: We checked the antigenic similarity of both viruses by comparing the amino acid sequence homology. The accession numbers of the NA of CA/09 and DH/07 have been included in the materials and methods (Line 122-124). The amino acid identity is indicated in the results (Line 324-325 and 333).
- In M & M, line 136: please add the title of (First experiment-Vaccination challenge study).
Response: The title has been edited (Line 140).
- In M & M, line 150: please add the title of (Second experiment-Vaccination study).
Response: The title has been edited (Line 155).
- I didn’t see a negative control mice groups (1st study: non-vaccinated, non-challenged) (2ndstudy: non-vaccinated control), if you have these data, please add it.
Response: Thank you for your comment. In the first experiment, unfortunately, we did not include the non-vaccinated, non-challenged group. We will consider this in future experiments.
For the second experiment, the PBS (the vehicle) inoculated group served as the non-vaccinated control group.
Round 2
Reviewer 1 Report
All the comments raised earlier are addressed.
Reviewer 2 Report
Authors have addressed concerns regarding the manuscript.